# Effect of a Polyglycolic Acid Mesh Sheet (Neoveil™) in Thyroid Cancer Surgery: A Prospective Randomized Controlled Trial

**DOI:** 10.3390/cancers14163901

**Published:** 2022-08-12

**Authors:** Seong Hoon Kim, Jong Hyuk Ahn, Hye Jeong Yoon, Jae Hwan Kim, Young Mi Hwang, Yun Suk Choi, Jin Wook Yi

**Affiliations:** 1Department of Surgery, Inha University Hospital & College of Medicine, Incheon 22332, Korea; 2Department of Surgery, Chungbuk National University Hospital, Cheongju 28644, Korea; 3Department of Surgery, Incheon Baek Hospital, Incheon 22534, Korea

**Keywords:** thyroidectomy, lymph node dissection, seroma, chyle, polyglycolic acid

## Abstract

**Simple Summary:**

Papillary thyroid cancer (PTC) is the most common type of thyroid cancer. Surgery for PTC involves resection of the thyroid gland and lymph node dissection around thyroid. Lymph node dissection is associated with an increased amount of fluid from the dissection area and chyle leakage due to thoracic duct injury. A polyglycolic acid mesh sheet (Neoveil™) has been proven to prevent postoperative fluid leakage in other surgeries. So, we aim to evaluate whether Neoveil™ can reduce postoperative drainage and chyle leakage in surgery for PTC. With the use of Neoveil™, the amount of drainage significantly decreased on the postoperative 2nd day and postoperative total drainage amount was lower. Triglyceride level was lower in the Neoveil™ group but was not statistically significant. No adverse effect from the Neoveil™ was observed during 9 months follow up. Our study suggests that Neoveil™ can be applied to reduce postoperative drainage in thyroid surgery for PTC.

**Abstract:**

Papillary thyroid cancer (PTC) is the most common type of thyroid cancer. Surgery for PTC involves resection of the thyroid gland and central lymph node dissection. Central lymph node dissection is associated with an increased amount of fluid from the dissection area and chyle leakage due to thoracic duct injury. There are few studies that deal with reducing fluid drainage and preventing chyle leakage after thyroid surgery with central lymph node dissection. A polyglycolic acid mesh sheet (Neoveil™) has been demonstrated to prevent postoperative fluid leakage in other surgeries. This study aims to evaluate whether a polyglycolic acid mesh sheet can reduce postoperative drainage and chyle leakage in papillary thyroid cancer surgery, and this study was designed as a prospective, open-label, randomized controlled trial in a single university hospital. The patients were randomly assigned to having only fibrin glue used in the central node dissection area (control group) or to having a polyglycolic acid mesh sheet applied after fibrin glue (treatment group). A total of 330 patients were enrolled, of which 5 patients were excluded. A total of 161 patients were included in the treatment group, and 164 patients were included in the control group. The primary outcome was the drainage amount from the Jackson-Pratt drain, and the secondary outcome was the triglyceride level in the drained fluid on the 1st and 2nd postoperative days. The drainage amount was significantly lower in the treatment group on the 2nd postoperative day (60.9 ± 34.9 mL vs. 72.3 ± 38.0 mL, *p* = 0.005). The sum of drainage amount during the whole postoperative days (1st and 2nd days) was also significantly lower in the treatment group (142.7 ± 71.0 mL vs. 162.5 ± 71.5 mL, *p* = 0.013). The postoperative triglyceride levels were lower in the treatment group but were not statistically significant (92.1 ± 60.1 mg/dL vs. 81.3 ± 58.7 mg/dL on postoperative day 1, *p* = 0.104 and 67.6 ± 99.2 mg/dL vs. 53.6 ± 80.4 mg/dL on postoperative day 2, *p* = 0.162). No adverse effects were observed in the treatment groups during the postoperative 9-month follow-up. Our study suggests that polyglycolic acid mesh sheets can be safely applied to reduce postoperative drainage amount in thyroidectomy patients who need lymph node dissection.

## 1. Introduction

Thyroid cancer is among the most common solid organ cancers worldwide [1]. According to the national statistics in South Korea, more than 30,000 thyroid surgeries are performed in a year [2]. The most important treatment for thyroid cancer is surgical resection of the involved thyroid gland [3]. Among the many subtypes of thyroid cancer, papillary thyroid cancer (PTC) accounts for the largest proportion, at approximately 90% [4,5]. Because lymph node metastasis is common in PTC, neck lymph node dissection, either prophylactically or therapeutically, is recommended in PTC surgery [3,6].

Various surgical complications can occur by thyroid gland resection, including voice problems due to recurrent laryngeal nerve injury and hypocalcemia by parathyroid injury. The complications associated with cervical lymph node dissection are seroma formation due to the increased amount of exudate from the node dissection area and chyle leakage by a thoracic duct injury, especially from left neck node dissection. The incidence of seroma and chyle leakage after thyroid surgery has been reported to be 4–6% in previous studies [7,8]. Although seroma and chyle leakage are not major complications after thyroid surgery, the hospitalization period may be extended, and continuous outpatient treatment may be needed. However, there is no study showing that proper substances or medical materials can reduce exudates and chyle leakage after thyroid surgery with neck lymph node dissection.

A polyglycolic acid mesh sheet (Neoveil™) is a tissue-strengthening repair agent that prevents air or fluid leakage after surgery. The preventive effect of polyglycolic acid mesh sheets has been proven in various surgical fields; these sheets diminish air leakage after lung resection surgery [9,10,11,12], reduce the pancreatic fistula incidence in pancreas resection [13,14,15,16], prevent bile leakage and hemorrhage after liver resection [17], and prevent bowel content leakage after colon surgery [18]. However, there are no clinical studies on whether polyglycolic acid mesh sheets can reduce exudates from the neck node dissection area and have a protective effect against chyle in thyroid surgery. The aim of this study was to investigate the effect of a polyglycolic acid mesh sheet in thyroid cancer surgery with lymph node dissection on exudative drainage and chyle leakage using a randomized controlled trial.

## 2. Materials and Methods

### 2.1. Patients

This study was designed as a prospective, open-label, randomized, controlled trial at a single university hospital. Patients aged 20–70 years who were diagnosed with PTC or suspicious PTC by fine-needle aspiration cytology and scheduled for thyroidectomy with neck lymph node dissection were enrolled in this study. The inclusion criteria were as follows: tumors without perithyroidal organ infiltration, no clinical evidence of distant organ metastasis, normal vocal cord function on laryngoscopic or laryngeal ultrasound prior to surgery, and no significant abnormal laboratory findings before surgery.

The exclusion criteria were as follows: patients who had taken aspirin or antiplatelet medication within 7 days before surgery, severe medical comorbidities (uncontrolled hypertension or diabetes, chronic renal failure, coagulation disorder), cardiovascular disease (Angina, heart failure, myocardial infarction, history of coronary artery disease, stroke, transient ischemic attack), drug or alcohol abuse, history of esophagus or airway disease, previous history of neck radiation or neck surgery, allergic history that needed medical treatment, participation in other clinical trials without 1 month, and pregnant or lactating women. All data were collected at the author’s institution (Inha University Hospital, Incheon, Korea).

The ethics of this study were approved by the institutional review board of the author’s institution (INHAUH 2018-10-008). This study has been listed on ClinicalTrials.gov since 25 April 2018 (NCT03510143). All enrolled patients provided written informed consent about this study protocol before the surgery day.

### 2.2. Randomization and Surgical Procedures

After obtaining informed consent from the patients, they were randomly assigned to two groups: treatment and control. All surgeries were performed by a single endocrine surgeon (JW Yi). The steps for thyroid surgery were performed as follows: Under general anesthesia, a 5–7 cm transverse skin incision and subcutaneous flap elevation were performed. The strap muscle was divided to obtain exposure of the thyroid gland. The isthmus was cut, and thyroid upper pole dissection was performed with superior thyroidal vessel ligation. Dissection of the lateral aspect of thyroid was performed with middle thyroidal vein ligation, the recurrent laryngeal nerve (RLN) was identified, and the inferior thyroidal vessel was ligated. The cervical lymph nodes were dissected either prophylactically or therapeutically. Before surgical wound closure, only fibrin glue (greenplast™) was applied to the lymph node dissection area in the control group, as shown in Figure 1A. In the treatment group, a polyglycolic acid mesh sheet additionally covered the lymph node dissection area after fibrin glue was applied, as shown in Figure 1B. A Jackson-Pratt (JP) drain was placed in the thyroidectomy bed in all patients.

### 2.3. Outcome Evaluation

The primary endpoints were the 24 h drainage amount (mL/day) from the JP drain and the triglyceride level (mg/dL) in the drained fluid on the 1st and 2nd postoperative days. The baseline time was every 6:00 a.m. on the postoperative days. Removal of the JP drain was permitted on the 3rd postoperative day. The triglyceride level in the drained fluid was measured at 7:00 a.m. on the 1st and 2nd postoperative days. The secondary endpoints are adverse events and safety assessments according to the polyglycolic acid mesh sheet during the admission days in the outpatient clinic at 2 weeks, 3 months, and 9 months postoperatively. Vocal cord evaluation was performed by laryngoscopic examination or laryngeal ultrasound at the outpatient clinic. Vocal cord palsy within 2 weeks of surgery was defined as transient and was defined as permanent 6 months after surgery. Transient hypoparathyroidism was defined as an intact parathyroid hormone (iPTH) level below 5 pg/mL within 2 weeks of surgery. An iPTH level below 10 pg/mL 6 months after surgery was defined as permanent hypoparathyroidism. The adverse events from the polyglycolic acid sheet included infection, irritation, allergic reaction and shock. The adverse events were checked at 2 weeks, 3 months and 9 months postoperatively at the outpatient clinic by the clinician.

### 2.4. Sample Size Calculation

At the time of our study (July 2019), there was no clinical study that used a polyglycolic acid mesh sheet in thyroid surgery. As such, the sample size was calculated from a previous study in thyroid surgery, according to the use of fibrin sealant. They compared postoperative drainage and triglyceride levels after thyroid surgery [19]. A total of 78 patients were compared between the fibrin sealant-treated group and the nontreated group. The sum of drain amounts was 93.5 ± 30.7 mL in the fibrin sealant group and 105.7 ± 31.2 mL in the control group (*t* test *p* value = 0.05). Using this result, we set the alpha error as 0.08 and the 1-beta error as 0.95. Finally, a total of 300 study subjects were required in our study according to the sample size calculation using G*Power software version 3.1.9.2 (Bonn, Germany). Considering the dropout rate of approximately 10%, the number of subjects required for recruitment was set up to 165 in each group (a total of 330 subjects).

### 2.5. Statistics

Patients were allocated to either the treatment or control group in a 1:1 ratio using a randomization program encoded by the R-program that was encoded by the corresponding author. After obtaining informed consent, randomization was performed on the day before surgery by the senior researcher (YM Hwang). Blinding was only applied to the patients who did not know their allocation status until the completion of the 9-month follow-up. An unpaired *t* test was used to compare the continuous variables. *Chi*-square or Fisher’s exact test was applied to the cross-table analysis according to the sample size. All statistical analyses were conducted using the R programming language, version 4.0.5. (R: A language and environment for statistical computing. Vienna, Austria. https://www.R-project.org/ (accessed on 31 March 2022)).

## 3. Results

From July 2019 to April 2021, a total of 330 patients were enrolled. Of these, five patients were excluded from the analysis. One patient denied continuing this study, and four patients were lost to follow-up due to the patients’ personal situations. Finally, 161 patients were included in the treatment group, and 164 patients were analyzed in the control group (Figure 2).

Table 1 shows the clinical characteristics of the 325 enrolled patients. A total of 253 were women, and 72 were men. Age, sex and body mass index (BMI) were not different between the two groups. In the preoperative fine-needle aspiration cytology, 225 patients had PTC (Bethesda category VI), and 100 patients had suspicious PTC (Bethesda category V) and planned to have neck lymph node dissection. The main tumor locations were not different between the two groups. A total of 140 lobectomies and 185 total thyroidectomies were performed. Central lymph node dissection (neck node level 6) was performed in 311 patients, and central plus lateral neck node dissection was performed in 14 patients. Operation time, estimated blood loss, and postoperative hospital stay days were not different between the two groups.

The pathological findings are described in Table 2. The pathological findings were PTC in 315 patients, nodular hyperplasia in 5 patients, noninvasive follicular thyroid neoplasm with papillary-like nuclear features in 3 patients, etc. There were no significant differences between the two groups in the patient pathology, tumor size, proportion of gross extrathyroidal extension, presence of lymph node metastasis, or the number of metastatic lymph nodes and harvested lymph nodes, as described in the table.

The postoperative outcomes are listed in Table 3. There was no significant difference in the drainage amount on the first postoperative day between the control and treatment groups (90.2 ± 43.5 mL vs. 81.8 ± 44.4 mL, *p* = 0.085). However, the drainage amount was significantly lower in the treatment group on the 2nd postoperative day (60.9 ± 34.9 mL vs. 72.3 ± 38.0 mL, *p* = 0.005). The total drainage amount after surgery was also significantly lower in the treatment group (142.7 ± 71.0 mL vs. 162.5 ± 71.5 mL, *p* = 0.013). Triglycerides on postoperative days 1 and 2 were lower in the treatment group than in the control group, but there was no statistical significance (81.3 ± 58.7 mg/dL vs. 92.1 ± 60.1 mg/dL, *p* = 0.104 on postoperative day 1; 53.6 ± 80.4 mg/dL vs. 67.6 ± 99.2 mg/dL, *p* = 0.162 on postoperative day 2). The incidence of seroma was higher in the control group (9 vs. 3 cases), but the difference was not statistically significant. Chyle leakage occurred in two patients: one in the control group and the other in the treatment group. Representative complications according to thyroidectomy, hyperparathyroidism and vocal cord palsy were also not statistically significant. During the 9-month follow-up after surgery, no adverse effects related to the polyglycolic acid mesh sheet were observed in the treatment group.

## 4. Discussion

Although PTC grows slowly and has a favorable survival outcome, cervical lymph node metastasis is very common. The rate of central lymph node metastasis in PTC was reported to be 16.9–53.5% in previous studies [20,21]. The 2015 American Thyroid Association (ATA) guideline recommended therapeutic central compartment lymph node dissection for patients with clinically involved central node metastasis, accompanied by total thyroidectomy. Prophylactic central-compartment neck dissection is considered in patients with PTC that do not clinically involve the central neck lymph nodes and who have advanced primary tumors or lateral neck nodes that are clinically involved, or if the information will be used to plan further steps in therapy. In Korea, when lymph node metastasis is found in the final pathology after surgery, it is not easy to convince the patient that there is no problem in survival or recurrence even if there is lymph node metastasis, and sometimes it leads to legal disputes. Therefore, our hospital’s policy is to perform prophylactic central node dissection (CND) for Bethesda V and VI patients and check frozen biopsy for lymph nodes. If lymph node metastasis is discovered during surgery, total resection may be performed depending on the size or proportion of the metastasized lymph node.

After thyroid surgery, exudative fluid is produced at the thyroidectomy site and lymph node dissection area. This exudate comes out naturally during the wound healing process, will gradually decrease if the amount is not large and will be absorbed into the body naturally [22]. However, if the exudate amount is too large, the exudate does not disappear and is retained, causing swelling of the surgery area and causing a seroma. Although a seroma is not a serious life-threatening complication, it is necessary to drain it with an appropriate procedure, such as aspiration or surgical drainage. If the seroma is not treated properly, a fibrous mass may remain in the surgery area, which could cause adhesions in the neck or cause an infection to develop [22,23,24]. To prevent these fluid-related complications, our hospital placed a Jackson-Pratt (JP) drain in all thyroid surgeries and maintained it until the amount of drainage was sufficiently reduced. Generally, on the 3rd postoperative day, we removed the JP drain when the daily drain amount reached under 50 mL/day.

In the deep neck area, the thoracic ducts were located. The thoracic duct carries chyle that contain both lymph and emulsified lipids. The thoracic duct starts from the 12th thoracic vertebra and extends to the neck [25]. During neck node dissection in the deep central neck (levels 6 and 7) and lateral neck (level 4), part of the thoracic duct can be injured, and chyle leakage may occur. A large amount of chyle is ejected from the surgical site and causes wound problems, infection, electrolyte imbalance and chylothorax [26,27]. The treatment of chyle leakage is mainly conservative, including a fat restriction diet or nil per os with total parenteral nutrition. However, repair surgery is required in severe cases [28]. The diagnosis of chyle leakage was made by clinical examination, and the triglyceride level in the drained fluid was checked. The incidence of chyle leakage after neck dissection is reported to be very low [7]. As such, the triglyceride level is an alternative reference for chyle leakage in clinical studies.

Many studies have been conducted to reduce complications after thyroid surgery, but most of them have focused on recurrent laryngeal nerve injury and hypoparathyroidism. There are few studies about reducing the exudate fluid or chyle leakage. Some studies have found that fibrin glue can reduce the drainage amount and prevent seroma formation [29,30]. A polyglycolic acid mesh sheet is a tissue-strengthening agent that reduces air or fluid leakage from the surgical site. Many clinical studies have been conducted, and they proved the preventive effect of polyglycolic acid mesh sheets for air, fluid and bowel content leakage [9,11,12,13,14,15,16,18]. However, no study has shown that the polyglycolic acid mesh sheet is effective in thyroid surgery for reducing exudate or chyle leakage. Therefore, we designed a randomized study to determine whether polyglycolic acid mesh sheets could reduce drainage and prevent chyle leakage after thyroid surgery. The manufacturer’s instructions for use recommend the use of polyglycolic acid mesh in combination with fibrin glue. Therefore, this study was conducted according to the guidelines. Future studies may be able to better prove the results using only polyglycolic acid mesh sheets.

According to our clinical trial, we found that the application of a polyglycolic acid mesh sheet reduced fluid drainage on the 2nd postoperative day and the total drainage amount (72.3 ± 38.0 mL vs. 60.9 ± 34.9 mL and 162.5 ± 71.5 mL vs. 142.7 ± 71.0 mL, respectively). The seroma formation rate was also low in the treatment group (9/164 in control vs. 3/161 in treatment). For chyle leakage, the triglyceride level was also lower on the 1st and 2nd postoperative days, although it did not reach statistical significance (92.1 ± 60.1 mg/dL vs. 81.3 ± 58.7 mg/dL, 67.6 ± 99.2 mg/dL vs. 53.6 ± 80.4 mg/dL). A clinical diagnosis of chyle leakage was only observed in 1 patient per group (2 patients total), who both received lateral neck node dissection of the left side. Other complications were not significantly different between the control and treatment groups, as shown in Table 3. During the 9-month follow-up period, there were no adverse complications in the treatment group, such as infection, allergic reaction, or other unexpected complications.

From our study, we suggest that a polyglycolic acid mesh sheet after thyroidectomy with lymph node dissection can be safely applied to reduce exudative drainage without any adverse effects from this material. For chyle leakage prevention, we only showed a tendency of lower triglyceride levels in the treatment group and the same rate of clinical chyle leakage. Perhaps this result is due to the low incidence of chyle leakage from the central neck node dissection. Future studies with a larger number of patients are needed, especially patients who have had surgeries with a higher risk of thoracic duct injury, such as lateral neck lymph node dissection (modified radical neck dissection, MRND). This is a product paid by health insurance in South Korea, and the price per unit is 48.34 USD for medium (M) size and 96.69 USD for large (L) size (exchange rate as of 29 July 2022). With insurance, the patient only pays 20%, and cancer patients only pay 5% in South Korea. Therefore, this product appears to be reasonably priced for use in low volume centers.

The limitations of our study are as follows. All surgeries were performed by an endocrine surgery specialized surgeon who had performed more than 2000 thyroid surgeries, which may why there was a low incidence of chyle leakage in our study. If such a study is conducted in a novice surgeon who is starting during his or her earliest surgical experience, a more significant difference can be seen. To accurately analyze the effect of Neoveil on chyle leakage in thyroid surgery, only lateral node dissection, especially left modified radical neck (MRND) dissection, should be compared. However, in our study, the number of patients who underwent left MRND was small, so we analyzed them together with patients who underwent central node dissection. Subgroup analysis was added as Appendix A. Precise research through multicenter studies is needed in the future. Another limitation is the study characteristics about the blinding. In studies on the use of specific materials in surgery, achieving double blinding, including by surgeons, is very difficult. Although this may not have had a significant impact, it should be pointed out as a limitation of our study. Another limitation is the surgical extent. The incidence of neck lymph node metastasis is determined by the extent of surgery. Lateral lymph node is included in the scope of surgery when metastasis is clinically confirmed in the preoperative examination, and it is difficult to know the exact incidence rate because these cases are relatively few. Therefore, the incidence varies each studies, and lateral lymph node metastasis has been reported in the range of 3.1% to 65.4% [31]. As such, we conducted this study mainly using central lymph node dissection. Therefore, if this study is conducted only on lateral cervical lymph node dissection patients with a multicenter study, we think that better results will be obtained. The other limitation is the difference in measurement time after surgery. In our hospital, most of the surgeries for thyroid patients are completed before 10 a.m. to 3 p.m. Therefore, in order to compare the amount of drain for the same time, we measured each 24 h from 6:00 the day after surgery and measured as postoperative day 1, 2, and 3 drainage amounts. Since the time the operation ends is different for each patient, it may act as a bias in measuring the drain amount and analyzing the results. Since we cannot control this part, in future studies, we need to measure the amount of drain from the end time for a more accurate analysis.

## 5. Conclusions

The application of a polyglycolic acid mesh sheet in thyroid surgery with lymph node dissection may be effective in reducing postoperative drainage. The triglyceride level showed only a tendency to be lower in the treatment group. No adverse effects were observed after the application of the polyglycolic acid mesh sheet. We suggest that polyglycolic acid mesh sheets be safely applied for thyroidectomy patients who need lymph node dissection. Further studies will be conducted in MRND patients to prove the effect of chyle leakage by the polyglycolic acid mesh sheet.

## Figures and Tables

**Figure 1 cancers-14-03901-f001:**
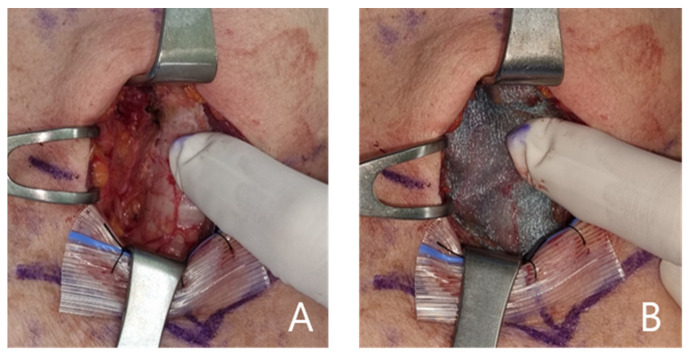
(**A**). Application of fibrin glue on the lymph node dissection site (control group). (**B**). Application of a polyglycolic acid mesh (Neoveil™) after applying fibrin glue on the lymph node dissection site (treatment group).

**Figure 2 cancers-14-03901-f002:**
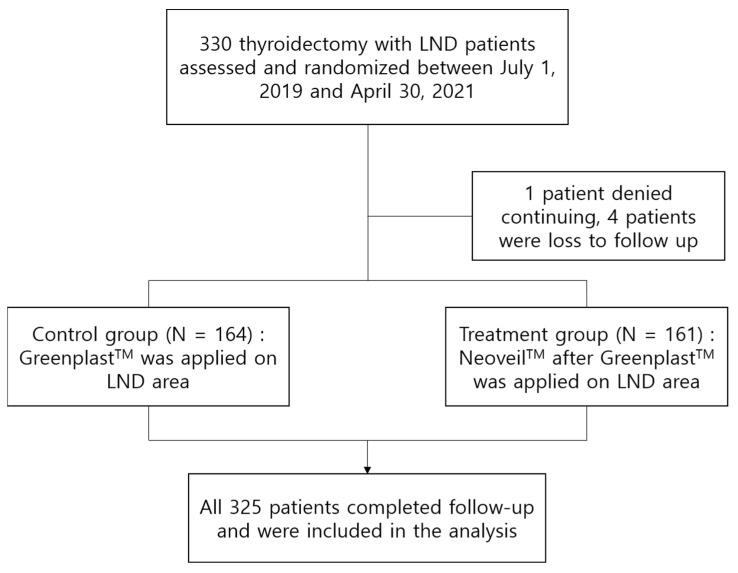
Flow chart for study enrollment.

**Table 1 cancers-14-03901-t001:** Clinical characteristics of the patients.

Variables	Control (n = 164)	Treatment (n = 161)	*p* Value
Age (years, mean ± sd)	45.6 ± 10.5	46.0 ± 11.7	0.722
Gender			
Women	127	126	0.859
Men	37	35	
Body mass index (BMI) (kg/m^2^, mean ± sd)	25.6 ± 4.1	25.3 ± 3.7	0.519
Fine-needle aspiration cytology			
Papillary thyroid cancer (VI)	119	106	0.189
Suspicious of papillary thyroid cancer (V)	45	55	
Tumor location			
Right	73	67	0.919
Left	62	62	
Isthmus	2	3	
Bilateral	27	29	
Thyroidectomy extent			
Lobectomy	67	73	0.414
Total thyroidectomy	97	88	
Lymph node dissection			
Central node dissection	157	154	0.972
Lateral node dissection	7	7	
Operation time (min, mean ± sd, 95% CI)	125.0 ± 40.9 (100.7–111.6)	123.3 ± 39.8 (100.7–110.5)	0.713
Estimated blood loss (mL, mean ± sd, 95% CI)	49.9 ± 76.9 (39.5–63.7)	46.6 ± 62.4 (37.5–56.9)	0.672
Hospital days after surgery (day, mean ± sd, 95% CI)	3.2 ± 0.9 (3.1–3.4)	3.3 ± 1.0 (3.1–3.4)	0.649

**Table 2 cancers-14-03901-t002:** Pathologic findings.

Variables	Control (n = 164)	Treatment (n = 161)	*p* Value
Pathologic diagnosis			0.189
Papillary thyroid cancer (PTC)	161	154	
Others *	3	7	
Largest tumor size (cm, mean ± sd, 95% CI)	1.1 ± 0.8 (0.9–1.2)	1.0 ± 0.7 (0.9–1.1)	0.295
Extrathyroidal extension (Gross)			
Absent	143	144	0.529
Present	21	17	
Lymph node metastasis			0.553
Absent	95	88	
Present	69	73	
Number of metastatic lymph nodes (mean ± sd, 95% CI)	1.5 ± 3.3 (1.1–2.1)	1.4 ± 2.5 (1.1–1.8)	0.724
Number of harvested lymph nodes (mean ± sd, 95% CI)	6.6 ± 7.5 (5.5–7.8)	6.2 ± 6.7 (5.2–7.3)	0.601

* Five with nodular hyperplasia, three with noninvasive follicular thyroid neoplasm with papillary-like nuclear feature (NIFTP), one with follicular variant PTC, and one with medullary thyroid carcinoma (MTC).

**Table 3 cancers-14-03901-t003:** Postoperative outcomes.

Variables	Control (n = 164)	Treatment (n = 161)	*p* Value
Drain amount (mL, mean ± sd, 95% CI)			
Postoperative day 1	90.2 ± 43.5 (84.0–96.9)	81.8 ± 44.4 (75.0–88.3)	0.085
Postoperative day 2	72.3 ± 38.0 (66.6–78.4)	60.9 ± 34.9 (55.6–66.3)	0.005
Total	162.5 ± 71.5 (152.3–173.7)	142.7 ± 71.0 (131.9–153.0)	0.013
Triglyceride (mg/dL, mean ± sd, 95% CI)			
Postoperative day 1	92.1 ± 60.1 (83.1–101.7)	81.3 ± 58.7 (72.7–90.8)	0.104
Postoperative day 2	67.6 ± 99.2 (55.1–83.7)	53.6 ± 80.4 (43.3–68.3)	0.162
Complications			
Seroma	9	3	0.072
Bleeding	0	1	NA
Wound problem	5	1	0.104
Chyle leakage	1	1	NA
Hypoparathyroidism (transient)	23	19	0.550
Hypoparathyroidism (permanent)	5	1	0.104
Vocal cord palsy (transient)	1	2	0.551
Vocal cord palsy (permanent)	0	0	NA

## Data Availability

The data presented in this study are available in this article and Appendix A.

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
