# Peer review of "Effect of a Polyglycolic Acid Mesh Sheet (Neoveil™) in Thyroid Cancer Surgery: A Prospective Randomized Controlled Trial"

_cancers, 2022, doi:10.3390/cancers14163901_

Round 1
Reviewer 1 Report
The authors present an interesting paper about the use of polyglycolic acid mesh sheet in thyroid cancer surgery. There are no previous reports in this field.
The study is in general well designed. Primary and secondary outcomes of the study are well identified, as well as, results are clearly exposed. Nevertheless, in my opinion there is a not negligible surgical detail in the enrollment of patients. It is difficult to understand if an amount of patients in both study groups had undergone concomitant central and lateral neck dissection. If so, please specify.
About discussion, my concerns are the following:
- In the discussion section some concepts are somewhat redundant (lines 205-213), as they are reported in the introduction section.
- Among limitations, the authors state at the line 286 that "we conducted the study mainly using central neck dissection". In my opinion, this put in evidence two problems:
a) From a surgical point of view a collection after central neck dissection, even if a lymphatic leakage is theoretically possible, is a seroma. In the lateral neck dissection, especially on the left, the risk of a lymphatic collection is more likely, even if it occurs very unfrequently. These two conditions have different and distinct causes and treatments. In my opinion, patients undergone central and lateral CND should be analyzed separately, or conversely, the paper should be focused on patients undergone TT and central neck dissection.
- b) As a consequence, levels of triglycerides and drainage amounts should be compared based on exact extension of surgery.
- The authors should discuss how much this new tool could influence the procedures in terms of costs. This allow evaluating use of this device also by surgeons working in low – volume centers.
In the conclusions section the authors state, "the application of a polyglycolic acid mesh sheet in thyroid surgery with lymph node dissection was effective in reducing the postoperative drainage". Data shows that there is a statistically significant reduction of drainage amount in the second postoperative day and in the total. Nevertheless, this statistically significant difference does not reflect a clinically relevant advantage. In my opinion based on data presented evidence for recommending polyglycolic acid mesh sheet seems weak and should be reconsidered.
Author Response
The authors present an interesting paper about the use of polyglycolic acid mesh sheet in thyroid cancer surgery. There are no previous reports in this field.
The study is in general well designed. Primary and secondary outcomes of the study are well identified, as well as, results are clearly exposed. Nevertheless, in my opinion there is a not negligible surgical detail in the enrollment of patients. It is difficult to understand if an amount of patients in both study groups had undergone concomitant central and lateral neck dissection. If so, please specify.
About discussion, my concerns are the following:
- In the discussion section some concepts are somewhat redundant (lines 205-213), as they are reported in the introduction section.
(Ans)
Thanks for your point. We deleted the duplicate contents.
- Among limitations, the authors state at the line 286 that "we conducted the study mainly using central neck dissection". In my opinion, this put in evidence two problems:
a) From a surgical point of view a collection after central neck dissection, even if a lymphatic leakage is theoretically possible, is a seroma. In the lateral neck dissection, especially on the left, the risk of a lymphatic collection is more likely, even if it occurs very unfrequently. These two conditions have different and distinct causes and treatments. In my opinion, patients undergone central and lateral CND should be analyzed separately, or conversely, the paper should be focused on patients undergone TT and central neck dissection.
b) As a consequence, levels of triglycerides and drainage amounts should be compared based on exact extension of surgery.
(Ans)
You are absolutely right and we agree with you. This is the point we were worried about when we started the study. In order to clearly analyse chyle leakage, only the lateral neck dissection, especially the left modified radical neck dissection, should be compared. As you know, because the number of lateral neck dissection is not large, a method such as a multicenter study is required to conduct a study. As you said, meaningful results have not yet been observed when only the lateral neck dissection is analyzed separately, so we added this to the limitation of the discussion as follows, and detailed analysis by surgical extent was included in the supplemental table.
(Add)
To accurately analyze the effect of Neoveil on chyle leakage in thyroid surgery, only lateral node dissection, especially left modified radical neck(MRND) dissection, should be compared. However, in our study, the number of patients who underwent left MRND was small, so we analyzed them together with patients who underwent central node dissection. Subgroup analysis was added as supplemental tables 1 and 2. Precise research through multicenter studies is needed in the future.
- The authors should discuss how much this new tool could influence the procedures in terms of costs. This allow evaluating use of this device also by surgeons working in low – volume centers.
(Ans)
Thank you for your suggestion, we added this contents in discussion as follow.
(Add)
This is a product paid by health insurance in Korea, and the price per unit is 48.34 USD for M size and 96.69 USD for L size. With insurance, the patient only pays 20%, and cancer patients pay 5% in South Korea. Therefore, this product appears to be reasonably priced for use in low volume centers.
In the conclusions section the authors state, "the application of a polyglycolic acid mesh sheet in thyroid surgery with lymph node dissection was effective in reducing the postoperative drainage". Data shows that there is a statistically significant reduction of drainage amount in the second postoperative day and in the total. Nevertheless, this statistically significant difference does not reflect a clinically relevant advantage. In my opinion based on data presented evidence for recommending polyglycolic acid mesh sheet seems weak and should be reconsidered.
(Ans)
As you said, we'll tone down our argument and amend it as follows:
(Before)
The application of a polyglycolic acid mesh sheet in thyroid surgery with lymph node dissection was effective in reducing the postoperative drainage.
(After)
The application of a polyglycolic acid mesh sheet in thyroid surgery with lymph node dissection may be effective in reducing the postoperative drainage.

Reviewer 2 Report
The authors conducted an RCT of over 300 patients with papillary thyroid cancer. All patients underwent a neck dissection of level VI as part of their treatment. The surgical bed was then dressed in either fibrin glue alone or with ‘Neoveil’ and a suction drain was left. They managed to show a reduction in drainage output in the study group compared to the control group.
The study is well conducted, and the authors should be praised for the efforts involved in orchestrating an RCT that involves surgery, hundreds of patients, and a long follow-up. I do have few comments:
Major comments:
1. A central neck dissection is not routinely recommended in PTC, especially when performing a lobectomy. The patients in the study had a mean tumor diameter of 1 cm, and if 2SD are added it means that 95% of all tumors resected were less than 3 cm. What was the rational to perform a total thyroidectomy rather than a lobectomy? The 2015 ATA guidelines’ recommendation 36 states that a ‘Prophylactic central-compartment neck dissection (ipsilateral or bilateral) should be considered in patients with PTC with clinically uninvolved central neck lymph nodes (cN0) who have advanced primary tumors (T3 or T4) or clinically involved lateral neck nodes (cN1b), or if the information will be used to plan further steps in therapy. (Weak recommendation, Low-quality evidence) and that a ‘Thyroidectomy without prophylactic CND is appropriate for small (T1 or T2), noninvasive, clinically node-negative PTC (cN0) and for most follicular cancers. (Strong recommendation, Moderate-quality evidence). The authors also cite a reference (Ahn et al., #5) that shows no advantage for a CND in a recent RCT. What was the rationale for this CND in this cohort? Eight patients did not have a cancer in their final pathology and underwent a CND (NIFTP and nodular hyperplasia).
2. The routine use of a fibrin glue or of a drain are also controversial. Can the authors please compare their results to data published in the literature to see the added benefit of this protocol?
3. The results are reported as means+/-SD, and p values. Since most results overlap in the study and control groups it would be interesting to see the 95%CI of both.
4. The baseline drainage out put was considered at 6:00 am the following morning. I can assume that some patients were operated almost 24 hours before that time while other were operated in the afternoon, that is, almost 12 hours before the baseline time. These differences can influence the drainage output. Do the authors know the patient distribution in the morning and afternoon surgeries between the two groups investigated?
5. Chyle leaks are a rare event in a CND. Since there were 2 cases of this complication, it is unsurprising that the leaks occurred to patients who underwent a lateral ND. However, their drainage output can skew the data for both groups, as means+/-SD are reported and not medians, or quartiles data.
6. The authors cite So YK et al. systematic review and meta-analysis from 2016 (reference 31). The inclusion criteria were studies that reported a neck dissection of any sort (“AND ((lateral OR “lateral compartment” OR “level II” OR “level III” OR “level IV” OR “level V”) AND (lymph node OR nodal OR lymphatic”). Therefore, the statement in line 285 is imprecise. The incidence of lateral ND in patients having PTC is not 20.9%. Studies that reported on PTC with no neck dissection were not included in that systematic review.
Minor comments:
1. Please present data with only one number after the decimal point.
2. Line 239: ‘nil per os’ should be with a lower case ‘N’
3. Line 251: ‘bower’. Is that a typo? ‘bowel’ maybe?
4. Line 294: Use less decisive words, such as ‘may be safely applied…”.
Author Response
The authors conducted an RCT of over 300 patients with papillary thyroid cancer. All patients underwent a neck dissection of level VI as part of their treatment. The surgical bed was then dressed in either fibrin glue alone or with ‘Neoveil’ and a suction drain was left. They managed to show a reduction in drainage output in the study group compared to the control group.
The study is well conducted, and the authors should be praised for the efforts involved in orchestrating an RCT that involves surgery, hundreds of patients, and a long follow-up. I do have few comments:
Major comments:
1. A central neck dissection is not routinely recommended in PTC, especially when performing a lobectomy. The patients in the study had a mean tumor diameter of 1 cm, and if 2SD are added it means that 95% of all tumors resected were less than 3 cm. What was the rational to perform a total thyroidectomy rather than a lobectomy? The 2015 ATA guidelines’ recommendation 36 states that a ‘Prophylactic central-compartment neck dissection (ipsilateral or bilateral) should be considered in patients with PTC with clinically uninvolved central neck lymph nodes (cN0) who have advanced primary tumors (T3 or T4) or clinically involved lateral neck nodes (cN1b), or if the information will be used to plan further steps in therapy. (Weak recommendation, Low-quality evidence) and that a ‘Thyroidectomy without prophylactic CND is appropriate for small (T1 or T2), noninvasive, clinically node-negative PTC (cN0) and for most follicular cancers. (Strong recommendation, Moderate-quality evidence). The authors also cite a reference (Ahn et al., #5) that shows no advantage for a CND in a recent RCT. What was the rationale for this CND in this cohort? Eight patients did not have a cancer in their final pathology and underwent a CND (NIFTP and nodular hyperplasia).
(Ans)
As you point out, the ATA guidelines recommend prophylactic CNDs at T3 and T4. However, in Korea, when lymph node metastasis is found in the final pathology after surgery, it is not easy to convince the patient that there is no problem in survival or recurrence even if there is lymph node metastasis, and sometimes it leads to legal disputes. Therefore, our hospital's policy is to do prophylactic CND for Bethesda V and VI patients and check frozen biopsy for lymph nodes. If lymph node metastasis is discovered during surgery, total resection may be performed depending on the size or proportion of the metastasized lymph node. I would like to think of this as a special situation limited to Korea. This content has been added to the discussion as follows.
(Add)
In Korea, when lymph node metastasis is found in the final pathology after surgery, it is not easy to convince the patient that there is no problem in survival or recurrence even if there is lymph node metastasis, and sometimes it leads to legal disputes. Therefore, our hospital's policy is to do prophylactic central node dissection (CND) for Bethesda V and VI patients and check frozen biopsy for lymph nodes. If lymph node metastasis is discovered during surgery, total resection may be performed depending on the size or proportion of the metastasized lymph node.
The routine use of a fibrin glue or of a drain are also controversial. Can the authors please compare their results to data published in the literature to see the added benefit of this protocol?
(Ans)
We wrote about fibrin glue in the discussion with references. (line 248-250)
Neoveil recommended the use of fibrin glue together in the manufacturer's instructions for use, so the study was conducted according to the manufacturer's instruction for use. There have been previous studies on fibrin glue in thyroid surgery. We added about this in discussion as follows.
As you said, the use of drain in thyroid surgery is controversial. Most hospitals in Korea, including ours, perform thyroid surgery on an inpatient basis and patients are discharged 2-3 days after surgery. By inserting a drain during surgery, additional aspiration can be prevented after surgery, and the possibility of additional infection is low because the drain is managed by medical staff in the hospital and removed when discharge.
(Add)
The manufacturer's instructions for use recommend the use of polyglycolic acid mesh in combination with fibrin glue. Therefore, the study was conducted according to the guidelines. Future studies may be able to better prove the results using only polyglycolic acid mesh sheets.
3. The results are reported as means+/-SD, and p values. Since most results overlap in the study and control groups it would be interesting to see the 95%CI of both.
(Ans)
Thank you for your point, we added 95% CI in table.
4. The baseline drainage out put was considered at 6:00 am the following morning. I can assume that some patients were operated almost 24 hours before that time while other were operated in the afternoon, that is, almost 12 hours before the baseline time. These differences can influence the drainage output. Do the authors know the patient distribution in the morning and afternoon surgeries between the two groups investigated?
(Ans)
You are right, and in our hospital, most of the surgeries for thyroid patients are finished before 10am to 3pm. Therefore, in order to compare the amount of drain for the same time, we measured each 24 hours from 6:00 on the next day of surgery and measured as POD 1, POD 2 and POD 3 drain amount. For this part, we added it to the limitation.
(Add)
The other limitation is the difference in measurement time after surgery. In our hospital, most of the surgeries for thyroid patients are completed before 10 am to 3 pm. Therefore, in order to compare the amount of drain for the same time, we measured each 24 hours from 6:00 the day after surgery and measured as POD 1, POD 2, POD 3 drain amount. Since the time the operation ends is different for each patient, it may act as a bias in measuring the drain amount and analyzing the results. Since we cannot control this part, in future studies, we need to measure the amount of drain from the end time for a more accurate analysis.
Chyle leaks are a rare event in a CND. Since there were 2 cases of this complication, it is unsurprising that the leaks occurred to patients who underwent a lateral ND. However, their drainage output can skew the data for both groups, as means+/-SD are reported and not medians, or quartiles data.
(Ans)
Although chyle leakage occurred in 2 patients, the drain amount was less than 500 mL/day in both patients and recovered within 3-4 days. As pointed out in number 3, adding CI seems to be more helpful for comparison. Thanks for the good point.
The authors cite So YK et al. systematic review and meta-analysis from 2016 (reference 31). The inclusion criteria were studies that reported a neck dissection of any sort (“AND ((lateral OR “lateral compartment” OR “level II” OR “level III” OR “level IV” OR “level V”) AND (lymph node OR nodal OR lymphatic”). Therefore, the statement in line 285 is imprecise. The incidence of lateral ND in patients having PTC is not 20.9%. Studies that reported on PTC with no neck dissection were not included in that systematic review.
(Ans)
You are right. We had an error in the description. We edited the content.
(Before)
The incidence of lateral neck lymph node metastasis in PTC was reported to be 20.9% in previous literature
(After)
The incidence of neck lymph node metastasis is determined by the extent of surgery. Lateral lymph node is included in the scope of surgery when metastasis is clinically confirmed in the preoperative examination, and it is difficult to know the exact incidence rate because these cases are relatively few. Therefore, the incidence varies each studies, and lateral lymph node metastasis has been reported in the range of 3.1% to 65.4%.
Minor comments:
1. Please present data with only one number after the decimal point.
2. Line 239: ‘nil per os’ should be with a lower case ‘N’
3. Line 251: ‘bower’. Is that a typo? ‘bowel’ maybe?
4. Line 294: Use less decisive words, such as ‘may be safely applied…”.
(Ans)
We fixed our mistakes, thank you for the good point.

Round 2
Reviewer 1 Report
I appreciate efforts done in improving the manuscript. The authors have adequately clarified all points of interest, as well as some concerns about the study polpulation. I suggest publication in the present form.
Reviewer 2 Report
All comments were addressed by the authors.